# Inhibition of Protease-Activated Receptor-2 Activation in Parkinson’s Disease Using 1-Piperidin Propionic Acid

**DOI:** 10.3390/biomedicines12071623

**Published:** 2024-07-22

**Authors:** Santina Quarta, Michele Sandre, Mariagrazia Ruvoletto, Marta Campagnolo, Aron Emmi, Alessandra Biasiolo, Patrizia Pontisso, Angelo Antonini

**Affiliations:** 1Department of Medicine, University of Padova, 35122 Padova, Italy; santina.quarta@unipd.it (S.Q.); mariagrazia.ruvoletto@unipd.it (M.R.); alessandra.biasiolo@unipd.it (A.B.); 2Parkinson and Movement Disorders Unit, Padua Neuroscience Center (PNC), Center for Neurodegenerative Disease Research (CESNE), Department of Neuroscience, University of Padova, 35122 Padova, Italy; michele.sandre@gmail.com (M.S.); marta.campagnolo@unipd.it (M.C.); aron.emmi@unipd.it (A.E.); angelo.antonini@unipd.it (A.A.)

**Keywords:** neuroinflammation, microglia, neurodegenerative diseases, PAR2 inhibition

## Abstract

In Parkinson’s disease, neuroinflammation is a double-edged sword; when inflammation occurs it can have harmful effects, despite its important role in battling infections and healing tissue. Once triggered by microglia, astrocytes acquire a reactive state and shift from supporting the survival of neurons to causing their destruction. Activated microglia and Proteinase-activated receptor-2 (PAR2) are key points in the regulation of neuroinflammation. 1-Piperidin Propionic Acid (1-PPA) has been recently described as a novel inhibitor of PAR2. The aim of our study was to evaluate the effect of 1-PPA in neuroinflammation and microglial activation in Parkinson’s disease. Protein aggregates and PAR2 expression were analyzed using Thioflavin S assay and immunofluorescence in cultured human fibroblasts from Parkinson’s patients, treated or untreated with 1-PPA. A significant decrease in amyloid aggregates was observed after 1-PPA treatment in all patients. A parallel decrease in PAR2 expression, which was higher in sporadic Parkinson’s patients, was also observed both at the transcriptional and protein level. In addition, in mouse LPS-activated microglia, the inflammatory profile was significantly downregulated after 1-PPA treatment, with a remarkable decrease in IL-1β, IL-6, and TNF-α, together with a decreased expression of PAR2. In conclusion, 1-PPA determines the reduction in neuroglia inflammation and amyloid aggregates formation, suggesting that the pharmacological inhibition of PAR2 could be proposed as a novel strategy to control neuroinflammation.

## 1. Introduction

Parkinson’s disease (PD) is a prevalent neurodegenerative movement disorder, affecting 1% of people over 65 years of age, with this figure rising to 4% of the population by the age of 80 years (Kowal et al., 2013) [1]. This disease is characterized by the degeneration of brainstem nuclei and the accumulation of α−synuclein-containing aggregates (Lewy bodies and neurites) throughout the brain. An increased level of α-synuclein is associated with an increased expression of protease-activated receptor-2 (PAR2) in the brains of PD patients [2]. Neuroinflammation is considered a complex defense mechanism leading to neuronal cell death, but the factors involved in the inflammatory process are not completely understood. 

The hallmark sign of neuroinflammation in the brain is the presence of activated microglia. Microglia in the aged brain shows dystrophic morphology, the elevated expression of inflammatory markers, and the reduced expression of neuroprotective factors [3]. Microglial cells mediate communication between the Central Nervous System (CNS) and immune cells. This communication is impaired in the elderly, predisposing them to low-grade chronic inflammation and the onset of neurodegenerative diseases [4]. Microglial cells survey the local environment and clear cellular and/or bacterial debris via phagocytosis. These activities are both dependent on proteinases and their target receptors [5]. The involvement of PAR2 and serine proteases is particularly interesting in cases of dyskinesia, given the findings of altered trypsin and serpins expression [6]. PAR2 belongs to the family of protease-activated receptors (PARS) and all the four PARs are widely expressed in neurons and glial cells in the nervous system, regulating diverse cellular functions, including gene transcription, neuronal cell proliferation, differentiation, and survival [7,8]. In particular, PAR2 has been shown to have modulatory effects in the peripheral nervous system where the receptor plays important roles in inflammation, neuronal signaling, and nociception [9,10,11,12].

PAR2 has been linked with pro- and anti-inflammatory actions under normal conditions. However, in experimental models of inflammatory brain disease, the receptor determines mainly pro-inflammatory effects [13,14,15]. Similarly, the role of Toll-Like Receptor 4 (TLR4) in neurodegenerative diseases is not well understood. Amyloid β activates TLR4, induces neuroinflammation, and thereby drives neuronal apoptosis [16,17]. In contrast, it has been reported that the nonpyrogenic LPS-derived monophosphoryl lipid A (MPL) activates TLR4, while reducing the levels of neuroinflammation and improving Alzheimer’s disease-related pathology in vivo [18]. These opposite actions may reflect the dual role that microglia has in innate immune response in different conditions. When in surveillance mode, ramified microglia continuously monitor the environment through sampling the extracellular space via rapid, moving processes. Cellular debris and/or pathogens are detected and then removed by the microglia once it is activated and in phagocytotic mode [5,19]. Such activity depends on a complex interaction with proteinases and cellular adhesion molecules. This activity occurs in the immediate environment of the cellular process due to low concentrations of the proteinase and their tight control by irreversibly-acting serin protease inhibitors (serpins) [20], as uncontrolled proteolysis is an undesirable event. Serpins comprise a large superfamily of proteins which control processes that require tight regulation, such as blood coagulation, inflammation, and fibrinolysis [21]. The role of serpins in brain function is still unknown, although they have been implicated in multiple sclerosis (serpin-A5), Alzheimer’s disease (serpinA3), neuronal plasticity (neuroserpin) [22,23,24], and recently also in Parkinson’s disease (serpinG1) [25]. It is worth noting that recent data indicate that SerpinB3 is a key molecule for PAR2 synthesis and activation, which in turn determines increased levels of the SerpinB3-transcription factor CCAAT-enhancer-binding protein-β (C/EBP-β) in a positive loop manner [26]. The aim of our study was to evaluate the effect of 1-piperidine propionic acid (1-PPA), a recently identified allosteric inhibitor of PAR2 [27], in neuroinflammation and microglia activation in PD.

## 2. Materials and Methods

### 2.1. Isolation and Culture of Primary Human Fibroblasts 

We used primary skin fibroblasts of PD patients, since they are a useful model system for PD, as they present defined mutations of genes like PTEN-induced putative kinase 1 (PINK1) and Parkin which are involved in mitophagy and mitochondrial homeostasis, preventing mitochondrial stress-induced inflammation, oxidative stress, and cell death [28,29,30]. Human fibroblasts were obtained from skin biopsies of patients with Parkinson’s disease, with or without genetic mutations. Fibroblasts from the skin biopsy of two non- sufferers of Parkinson’s disease were used as control. Cells were grown in Dulbecco’s modified Eagle medium (DMEM) (Gibco) with the addition of 1% penicillin/streptomycin, 1% non-essential amino acids solution, 1 mM l-glutamine, and 10% FBS at 37 °C with 5% CO_2_ atmosphere. Preliminary experiments were conducted on primary human fibroblasts using increasing amounts of 1-PPA over a time frame of 0 to 72 h to identify the best reagent concentration and the optimal time point. On the basis of the results obtained, all the experiments described below were carried out in fibroblast cultures treated with 1-PPA at a concentration of 10 ng/mL (Merck Sigma-Aldrich St. Louis, MO, USA) for 48 h.

The study protocol received approval by the Ethical Committee for clinical experimentation of Padova Province (Prot. n. 0034435, 6 August 2020). Informed consent for the use of biological samples was obtained from all patients. All procedures on human tissue samples were carried out in accordance with the Declaration of Helsinki. 

### 2.2. Isolation and Culture of Mouse Microglia 

Primary mouse microglial cells were derived from mixed gender cultures of C57BL/6J (Charles River, Wilmington, MA, USA) mouse brains (postnatal days P0–P2). Cerebral cortices were stripped of the meninges and mechanically dissociated as previously described [31]. The cell suspension, obtained from two mouse brains, was plated on a poly-l-lysine (0.1 mg/mL, Merck Sigma-Aldrich, St. Louis, MO, USA)-coated T-75 flask and cultivated in DMEM, supplemented with 10% heat-inactivated fetal bovine serum and 1% penicillin/streptomycin (Gibco, Thermo Fischer Scientific, Waltham, MA, USA). The next day, the cells were washed three times with DPBS (Gibco, Thermo Fischer Scientific, Waltham, MA, USA) to remove cellular debris and cultured as previously reported [32]. After 8–10 days in culture, weakly attached mature microglial cells were detached from the astrocytic monolayer with a repetition of the harvesting procedure every 2–3 days, up to three times.

To induce microglia activation, cells were incubated with 100 ng/mL of LPS (Merck Sigma-Aldrich, St. Louis, MO, USA) for 24 h. The cells were then treated with 10 ng/mL of 1-PPA for 24 h, and the cell pellets were harvested and used for RNA extraction. 

### 2.3. Thioflavin S Staining

To detect amyloid-like structures, Thioflavin S (Th-S) staining was carried out, which led to an increase in specific fluorescence when excited under blue light; this can be easily monitored and quantified. The slides of primary human fibroblasts from PD patients were incubated with freshly prepared 0.05% Thioflavin S (Merck Sigma-Aldrich, St. Louis, MO, USA) for 8 min and quickly washed twice with 70% non-denatured ethanol, followed by three washes with H_2_O. Next, the slides were incubated with DAPI (1:1000 in methanol), mounted with Elvanol (Merck Sigma-Aldrich, St. Louis, MO, USA) and observed under a fluorescence microscope (Axiovert 200M-Apotome.2, Carl Zeiss MicroImaging GmbH, Göttingen, Germany). 

### 2.4. Immunofluorescence Analysis

Human primary skin fibroblasts were seeded on slides (4 × 10^5^ cells/slide); the day after they were incubated with 1-PPA (10 ng/mL) or with medium only for 48 h. 

Mouse microglial cells (6 × 10^5^ cells/slide) were activated using 100 ng/mL of LPS (Lipopolysaccharides from *Escherichia coli* O111:B4 Merck Sigma-Aldrich, St. Louis, MO, USA) for 24 h and treated with 10 ng/mL of 1-PPA or saline buffer as control for additional 24 h. 

Cells were fixed with 4% paraformaldehyde, permeabilized with 0.4% Tryton X-100, and blocked with 5% goat serum (Invitrogen Life Technologies, Waltham, MA, USA) in PBS containing 1% BSA. The slides were incubated with monoclonal anti-PAR2 antibody (1:1000) obtained in rabbit and anti-SB3 antibody (1:150) obtained in mice for 1 h at room temperature, followed by incubation with the Alexa-Goat 546 and 488 secondary antibodies (1:500), respectively. Cellular nuclei were counterstained with Dapi (Merck Sigma-Aldrich, St. Louis, MO, USA). Slides were mounted with Elvanol (Merck Sigma-Aldrich, St. Louis, MO, USA) and observed under a fluorescence microscope (Axiovert 200M-Apotome.2, Carl Zeiss MicroImaging GmbH, Göttingen, Germany).

### 2.5. Quantitative Real-Time PCR (Q-PCR) 

Total RNA was extracted from human fibroblasts and mouse microglial cells using Trizol Reagent (Invitrogen, Carlsbad, CA, USA) according to the manufacturer’s instructions. After the determination of the purity and the integrity of total RNA, complementary DNA synthesis was carried out from 1ug of RNA using LunaScript RT SuperMix (New England BioLabs, Ipswich, MA, USA). Quantitative real-time PCR reactions (RT-PCR) were performed according to the Luna Universal qPCR master Mix (New England Biolabs, Ipswich, MA, USA) protocol, using a CFX96 Real-Time instrument (Bio-Rad Laboratories Inc, Hercules, CA, USA). The relative gene expression was generated for each sample by calculating 2^-ΔCt^ [33].

Primers sequences used in the study are reported in Appendix A.

### 2.6. Statistical Analysis

Statistical analysis was performed using Student’s t-test for analysis of variance. All reported *p*-values were two-tailed and considered significant if *p* ≤ 0.05. The data in the bar charts are presented as mean ± SEM and were obtained from at least three independent experiments and at least 5 measurements/slides for immunofluorescence quantification. Statistical tests and the concentration–time curves were performed using GraphPad Prism version 6.07 for Windows (GraphPad Software, La Jolla, CA, USA). 

## 3. Results

### 3.1. Human Primary Fibroblasts

To investigate the role of 1-PPA in neurodegenerative diseases, we have used two in vitro models, namely human primary fibroblasts obtained from patients with PD and LPS-activated primary mouse microglial cells.

For the first model, it has been widely demonstrated that patient skin fibroblasts can be used as model systems for Parkinson’s disease, since amyloid aggregates are also detectable in this compartment and easily obtainable from skin biopsy [28,34,35]. In the fibroblast of all patients, including both genetic and sporadic PD, amyloid aggregates were visible using the Th-S assay, and treatment with 1-PPA for 48 h determined a decrease in fluorescence signal in all of the cases, reaching a significant difference in the fibroblasts of the two sporadic patients (Figure 1). 

It is worth noting that the same primary fibroblasts from patients with PD, when treated with 1-PPA in the conditions described above, documented a parallel decrease in expression not only of PAR2, but also of SerpinB3, which was previously found to play a relevant role in PAR2 synthesis and expression [26]. These results were obtained in the fibroblast from all patients, both with sporadic (Figure 2) and genetic PD (Figure 3).

While in untreated fibroblasts from sporadic PD patients the levels of PAR2 and SerpinB3 were higher than those observed in the controls, in genetic PD patients, the levels of these two molecules were similar to those observed in the controls. These results suggest that the SerpinB3/PAR2 axis is more involved in amyloid aggregates formation in patients with sporadic PD than in patients with genetic PD, although the decrease in these two molecules was determined using 1-PPA which was also found in the controls, ultimately favoring the reduction in amyloid deposition in all patients.

The above described data have also been confirmed at the level of gene expression where a decline of both PAR2 (Figure 4A) and of SerpinB3 (Figure 4B) after 1-PPA treatment was observed in all the patients.

### 3.2. Primary Mouse Microglia

To investigate the role of this novel compound in the context of the brain microenvironment, focusing on the inflammatory response of microglia, primary mouse microglial cells were isolated from C57BL/6J mouse brains, activated with 100 ng/mL of LPS, and then treated with 10 ng/mL of 1-PPA or with medium alone for 24 h. The expression of the analyzed inflammatory cytokines documented a significant downregulation of IL-1β, IL-6, and TNF-α in cells treated with increasing amounts of 1-PPA, compared to untreated cells (Figure 5).

In agreement with the above results, LPS-activated microglial cells treated with 1-PPA showed a significant decrease in PAR2 expression, both at protein and transcription level (Figure 6).

## 4. Discussion

In the neurodegenerative disorders, truly effective treatments are rare, and the socio-economic costs are expected to increase considerably over the next few decades [36]. Given the growing demand for central nervous system (CNS) drugs for an aging population, there is an urgent need for effective strategies to improve the success rates in drug discovery and development [37].

This study aimed to evaluate whether the novel compound 1-PPA, that we have recently demonstrated to inhibit PAR2 activation [27] and to markedly reduce the inflammatory response [26], could be an effective compound to reduce neuroinflammation in PD. Our findings have shown that this small molecule is able to reduce the number of amyloid aggregates, typically found in neurodegenerative diseases, in primary fibroblast cultures from patients with PD, independently of the presence of genetic mutations. This effect was corroborated via the analysis of PAR2 expression, which was reduced in the fibroblasts of all treated patients. It is interesting to note that in the untreated fibroblasts of patients with sporadic PD, the expression of PAR2 was higher than in the controls, while it varied with the fibroblasts of patients with genetic PD, although in both groups of patients and also in the controls 1-PPA treatment determined a decrease in the expression of this receptor. This behavior may suggest that PAR2 could play a primary role in amyloid formation in sporadic PD. However, also in genetic PD, the reduced expression of PAR2 below the physiological level could favor amyloid regression, with genetic mutation being the main driving mechanism in this case. The dual role of PAR2 in PD has been widely demonstrated. Indeed, from the one hand, this molecule can exert neuroprotective effects, especially in ischemic damage. On the other hand, the ability to block this molecule during the neuroinflammation process can slow down the neurodegenerative process, which is typical of these diseases [13].

Since it has been recently shown that the anti-protease activity of SerpinB3 exerts an important role in PAR2 activation and increased synthesis in metabolic-associated liver disease [26], we have assessed whether this also occurs in our model of neurodegenerative diseases. The obtained results have demonstrated the parallel behavior of PAR2 and SerpinB3 molecules in fibroblast cultures of all patients, with a decrease in both molecules after treatment with PAR2. These findings support the role of the PAR2/SerpinB3 axis in inflammation and provide evidence of its implication in amyloid deposition. The mechanism through which 1-PPA determines amyloid aggregation inhibition is likely due to SerpinB3 downregulation, following PAR2 inhibition, since this serpin determines the disfunction of the ubiquitin–proteasome system, preventing protein degradation via selective NEDDylation since it determines the over-expression of the NEDD8-activating enzyme 1 (NAE1) [38].

In the literature, it is well known that in neurological disorders the role of microglia is crucial, as it plays a defensive function; however, its persistent activation is one of the most important factors leading to neurodegeneration [39]. In our experimental conditions, using LPS-activated mouse microglia, treatment with 1-PPA induced a significant downregulation of PAR2 which was associated with a significant decrease in inflammatory cytokines synthesis, providing evidence that this small molecule can control brain inflammation.

These results are of particular relevance, since our preliminary data indicate that 1-PPA is able to cross the blood–brain barrier (manuscript in preparation), and this barrier greatly restricts and controls the movement of substances’ entry into the brain; brain drug delivery is one of the main problems for the treatment of neurological disorders [40].

## 5. Conclusions

Targeting PAR2, a molecule involved in many inflammatory diseases as well as in cancer, is an important and valuable strategy in combatting neurodegenerative disease, particularly in PD. In this study, we have shown that PAR2 inhibition with the novel compound 1-PPA leads to a reduction in neuroglial inflammation and a decrease in amyloid aggregates, typical features of neurodegenerative processes.

In conclusion, 1-PPA, being part of the very small group of molecules that can reach the brain, provides a new, concrete strategy to control neuroinflammation, through the pharmacological inhibition of the inflammatory pathway that targets the PAR2/SerpinB3 axis.

## 6. Patents

Italian Patent Application N. 102022000014593 filed by the University of Padova on 12 July 2022; PTC/IB2023/057138 filed on 12 July 2023.

## Figures and Tables

**Figure 1 biomedicines-12-01623-f001:**
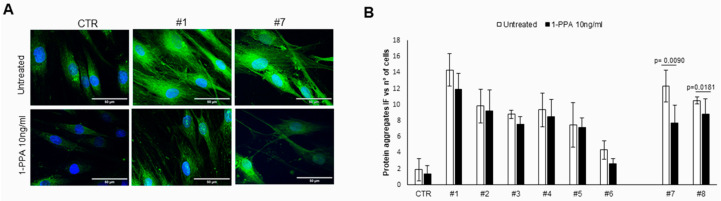
Amyloid aggregates in skin fibroblasts of patients with Parkinson’s disease. (**A**) Representative examples of amyloid aggregates detected using Thioflavine-S staining in human skin fibroblasts from a control (CTR) and from patients with Parkinson’s disease (genetic, #1 and sporadic, #7), untreated or treated with 10 ng/mL of 1-PPA for 48 h. (**B**) Quantification of immunofluorescence intensity of amyloid aggregates, obtained with the Zeiss AxioVision^®^ software (version 4.8), in the control (CTR) and in patients with genetic (#1–#6) and sporadic (#7 and #8) patients. Each value is expressed as the mean ± SEM of at least 5 measurements.

**Figure 2 biomedicines-12-01623-f002:**
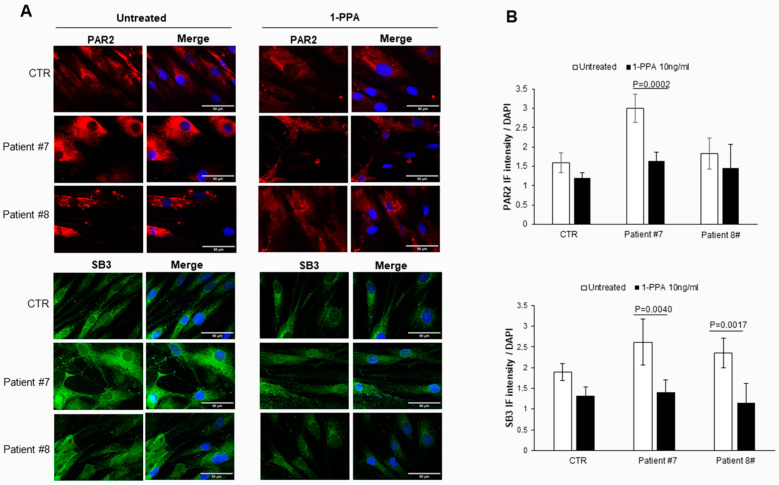
PAR-2 and SerpinB3 expression in skin fibroblasts of patients with sporadic Parkinson’s disease. (**A**) Immunofluorescence analysis of skin fibroblast of sporadic patients (#7 and #8) and of a representative control (CTR) untreated or incubated with 1-PPA (10 ng/mL) for 24 h. (**B**) Quantification of immunofluorescence intensity of PAR2 and SerpinB3, obtained using Zeiss AxioVision^®^ software, in skin fibroblasts untreated or treated with 1-PPA. Each value is expressed as the mean ± SEM of at least 5 measurements and the *p*-values were considered significant if *p* ≤ 0.05. Scale bar: 50 µm.

**Figure 3 biomedicines-12-01623-f003:**
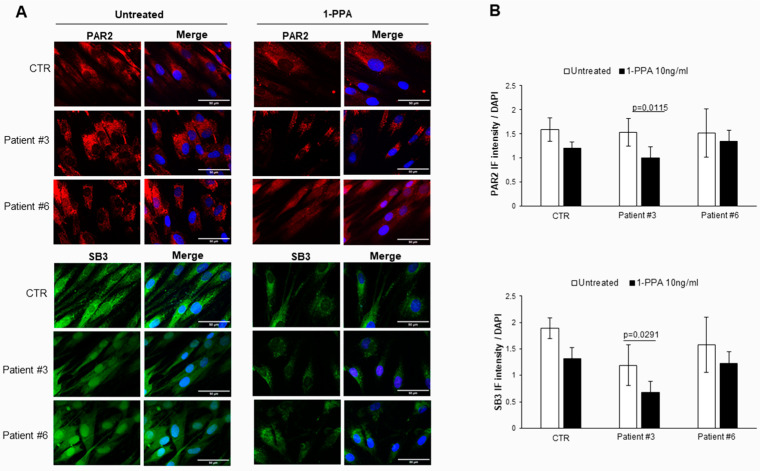
PAR2 and SerpinB3 expression in skin fibroblasts of patients with genetic Parkinson’s disease. (**A**) Representative examples of immunofluorescence analysis in patients with genetic Parkinson’s disease (#3 and #6) and in a control (CTR) in presence or not of 1-PPA (10 ng/mL) for 24 h. (**B**) Quantification of immunofluorescence intensity of PAR-2 and SerpinB3 expression after treatment with medium (CTR) or with 1-PPA, using Zeiss AxioVision^®^ software. Each value is expressed as the mean ± SEM of 5 measurements and the *p*-values were considered significant if *p* ≤ 0.05. Scale bar: 50 µm.

**Figure 4 biomedicines-12-01623-f004:**
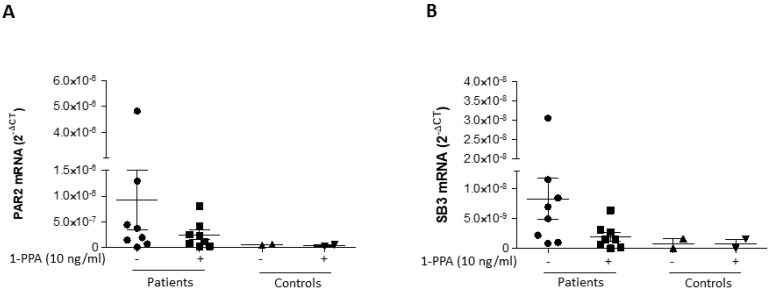
Gene expression of PAR2 and of SerpinB3 in skin fibroblasts of patients with Parkinson’s disease. Distribution of quantitative real-time PCR results of PAR2 (**A**) and of SerpinB3 (SB3) (**B**) mRNA in skin fibroblasts of all patients and of the controls, treated or untreated for 24 h with 10 ng/mL of 1-PPA. Results are expressed as 2^-ΔCT^ values; central bar represents mean ± SEM.

**Figure 5 biomedicines-12-01623-f005:**
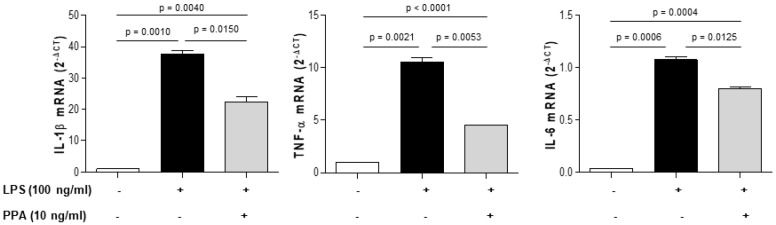
Effect of 1-PPA on cytokine expression in activated mouse microglia. Quantitative real-time PCR of inflammatory cytokine genes (IL-1β, TNF-α, IL-6) in primary mouse microglia activated with LPS 100 ng/mL for 24 h and treated or not with 1-PPA at 10 ng/mL concentration. Results are expressed as the mean ± SEM of 2^-ΔCT^ values of three different experiments.

**Figure 6 biomedicines-12-01623-f006:**
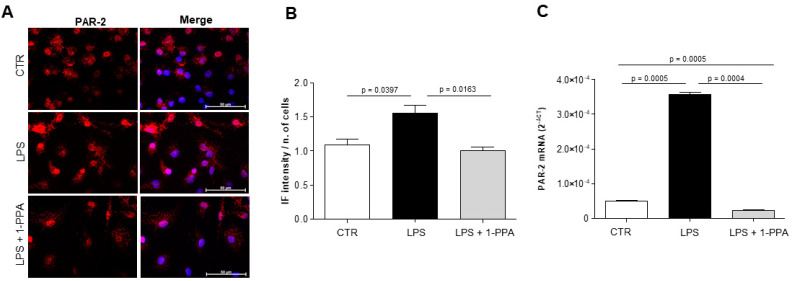
PAR2 expression in mouse microglia. (**A**) Representative examples of immunofluorescence analysis of PAR2 in primary mouse microglia cells (CTR) and in LPS-activated microglia in presence or in no presence of 1-PPA. (**B**) Quantification of immunofluorescence intensity of PAR2 expression in control cells and in LPS-activated cells untreated or treated with 1-PPA. Each value is expressed as the mean ± SEM of at least 5 measurements, using Zeiss AxioVision^®^ software. (**C**) Parallel analysis of quantitative real-time PCR of PAR2 mRNA in microglial cells and in LPS-activated cells untreated or treated with 10 ng/mL of 1-PPA for 24 h. Results are expressed 2^-ΔCT^. The *p*-values were considered significant if *p* ≤ 0.05.

## Data Availability

The data presented in this study are available upon request from the corresponding author.

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
