# Peer review of "Inhibition of Protease-Activated Receptor-2 Activation in Parkinson’s Disease Using 1-Piperidin Propionic Acid"

_biomedicines, 2024, doi:10.3390/biomedicines12071623_

Round 1

Reviewer 1 Report

Comments and Suggestions for Authors

Comments

MS # biomedicines-3113329

Title: INHIBITION OF PROTEASE ACTIVATED RECEPTOR-2 ACTIVATION IN
PARKINSON’S DISEASE

Overview: This study focused on the potential of 1-Piperidin Propionic Acid (1-PPA) as Proteinase-activated receptor-2 (PAR2) inhibitor with anti-inflammatory properties that could help treat Parkinson's disease.

In my opinion, this manuscript can be accepted for publication after Major Revision

I have highlighted my concerns in the manuscript.

Major comments

1.       As the study mentions about inhibition of PAR-2 inhibition by 1-Piperidin Propionic Acid (1-PPA), I suggest the authors include its name in the Title as well.

2.       Abstract: “In Parkinson’s…..defence mechanism” is misleading and downgrades the role of 1-PPA as an inflammation inhibitor. As neuroinflammation is a double-edged sword and does not always act as a defence mechanism. When inflammation occurs unnecessarily it can have harmful effects, despite its important role in battling infections and healing tissue. Once triggered by microglia, astrocytes transition into a reactive state and shift from supporting the survival of neurons to causing their destruction.

3.       Section 2.4: At what dilutions antibodies were used?

4.       Section 2.7 statistical analysis: “The data in the bar charts are presented as mean ± SEM and were obtained from at least three independent experiments.”

But in the figures authors mentioned, “Each value is expressed as the mean ± SD of at least 5 measurements.” So is it SEM or SD?

5.       Section 2.7 statistical analysis: “All reported p-values were two-tailed and considered significant if p ≥ 0.05” but your results say “p-values were considered significant if p ≤ 0.05.”

6.       Discussion: It would be better if the authors relate the structure of 1-PPA with amyloid aggregation inhibition. What are the structural characteristics to inhibit aggregation?

Minor

1.      Parkinson’s disease has already been mentioned as PD in line 33 hence the term “Parkinson’s disease” in the text can be replaced by PD.

2.      Figures: change “10ng/ml” to “10 ng/ml” in the figure and figure legend (and also in the text e.g. line 116,132)

3.      Line 260: Change “PAR2below” to “PAR2 below”

4.      Revise the axis title fonts and make them bold for better visualization.

5.      Please change 105 to 105 while mentioning the number of cells.

Comments on the Quality of English Language

Reviewer 2 Report

Comments and Suggestions for Authors

Manuscript ID: biomedicines-3113329

Title: INHIBITION OF PROTEASE ACTIVATED RECEPTOR-2 ACTIVATION IN PARKINSON’S DISEASE

The paper titled, " Inhibition of protease activated receptor-2 activation in Parkinson’s disease ", is well-written and well-organized. 

The authors have systematically presented the study.

The authors thoroughly reviewed the literature for writing this article.

The global message of the manuscript is clear.

The role of PAR2  in Parkinson’s disease and cancer has been under investigation and the information included in this paper can be a very important consideration for many investigators. I recommend the acceptance of this manuscript.

There is only one comment that needs to be addressed to the authors. In the section Materials and Methods please write the origin of LPS ( from what the serotype of bacteria is LPS) and describe the catalog number.

Round 2

Reviewer 1 Report

Comments and Suggestions for Authors

The suggested changes have been made by the authors. The manuscript can be accepted for the publication.